# Identification of Novel *phoP-phoQ* Regulated Genes that Contribute to Polymyxin B Tolerance in *Pseudomonas aeruginosa*

**DOI:** 10.3390/microorganisms9020344

**Published:** 2021-02-09

**Authors:** Baopeng Yang, Chang Liu, Xiaolei Pan, Weixin Fu, Zheng Fan, Yongxin Jin, Fang Bai, Zhihui Cheng, Weihui Wu

**Affiliations:** State Key Laboratory of Medicinal Chemical Biology, Key Laboratory of Molecular Microbiology and Technology of the Ministry of Education, Department of Microbiology, College of Life Sciences, Nankai University, Tianjin 300071, China; g8_1980530@outlook.com (B.Y.); liuchangbarry@gmail.com (C.L.); pxlay@hotmail.com (X.P.); 1120200520@mail.nankai.edu.cn (W.F.); jmy@xnmsn.com (Z.F.); yxjin@nankai.edu.cn (Y.J.); baifang1122@nankai.edu.cn (F.B.); zhihuicheng@nankai.edu.cn (Z.C.)

**Keywords:** polymyxin B, *Pseudomonas aeruginosa*, PhoP-PhoQ, membrane integrity

## Abstract

Polymyxin B and E (colistin) are the last resorts to treat multidrug-resistant Gram-negative pathogens. *Pseudomonas aeruginosa* is intrinsically resistant to a variety of antibiotics. The PhoP-PhoQ two-component regulatory system contributes to the resistance to polymyxins by regulating an *arnBCADTEF*-*pmrE* operon that encodes lipopolysaccharide modification enzymes. To identify additional PhoP-regulated genes that contribute to the tolerance to polymyxin B, we performed a chromatin immunoprecipitation sequencing (ChIP-Seq) assay and found novel PhoP binding sites on the chromosome. We further verified that PhoP directly controls the expression of PA14_46900, PA14_50740 and PA14_52340, and the operons of PA14_11970-PA14_11960 and PA14_52350-PA14_52370. Our results demonstrated that mutation of PA14_46900 increased the bacterial binding and susceptibility to polymyxin B. Meanwhile, mutation of PA14_11960 (*papP*), PA14_11970 (*mpl*), PA14_50740 (*slyB*), PA14_52350 (*ppgS*), and PA14_52370 (*ppgH*) reduced the bacterial survival rates and increased ethidium bromide influx under polymyxin B or Sodium dodecyl sulfate (SDS) treatment, indicating roles of these genes in maintaining membrane integrity in response to the stresses. By 1-N-phenylnaphthylamine (NPN) and propidium iodide (PI) staining assay, we found that *papP* and *slyB* are involved in maintaining outer membrane integrity, and *mpl* and *ppgS*-*ppgH* are involved in maintaining inner membrane integrity. Overall, our results reveal novel PhoP-PhoQ regulated genes that contribute to polymyxin B tolerance.

## 1. Introduction

*Pseudomonas aeruginosa* is an opportunistic Gram-negative bacterial pathogen that causes various infections in immunocompromised patients, including those suffering from acquired immune deficiency syndrome (AIDS), cancer, burn injury, and cystic fibrosis (CF) [1,2]. *P. aeruginosa* is intrinsically resistant to a variety of antibiotics [3]. The resistant mechanisms include low membrane permeability, multidrug efflux systems as well as chromosome encoded or horizontally acquired antibiotic breakdown or modification enzymes [4,5,6]. In addition, formation of biofilm drastically enhances the bacterial resistance to antibiotics [7]. Polymyxin B and colistin (polymyxin E) are cyclic polypeptide antibiotics that are used as the last-resort treatment option against serious infections caused by Gram-negative bacteria, including *P. aeruginosa*, *Klebsiella pneumoniae*, and *Acinetobacter baumannii* [7,8,9,10]. Polymyxin B and colistin are very similar in their structures, both consisting of a cyclic heptapeptide, a linear tripeptide, and a fatty acid chain [11]. The only difference between polymyxin B and colistin is an amino acid in the heptapeptide ring, with a leucine in colistin and a phenylalanine in polymyxin B [12]. The hydrophilic and hydrophobic regions of polymyxin B and colistin are essential for their antimicrobial activity [13]. The positively-charged polymyxins bind to the negatively-charged phosphate groups of lipopolysaccharide (LPS) on the Gram-negative bacteria surface [14]. Then the fatty acid chain of the polymyxin is inserted into the bacterial outer membrane (OM), increasing the membrane permeability and facilitating the passage of the polymyxin through the outer membrane [15]. Subsequently, the polymyxin damages the integrity of the bacterial inner membrane (IM) [16]. In addition, polymyxins have been demonstrated to inhibit NADH-quinone oxidoreductases on the bacterial cell membrane and interfere with cell division [17].

In *P. aeruginosa*, the two-component regulatory system PhoP-PhoQ is involved in the bacterial resistance to polymyxins [18]. The *phoP*-*phoQ* genes forms an operon with an outer membrane porin gene *oprH* (*oprH*-*phoP*-*phpQ*), which is positively regulated by PhoP [19]. Under low-Mg^2+^ environments, the phosphorylated PhoP directly controls target gene expression by binding to a conserved sequence in the promoter region. The sensor protein PhoQ functions as a phosphatase that dephosphorylates the cognate response regulatory protein PhoP [20]. Thus, defect in the *phoQ* gene results in constitutive expression of the PhoP regulated genes [21]. PhoP directly binds to the promoter region of the *oprH*-*phoP*-*phoQ* operon and activates its transcription [19]. The high-level expression of OprH facilitates the uptake of divalent cations [22]. In addition to its own operon, PhoP has been found to regulate multiple genes that are involved in transmembrane transport, LPS modification, antibiotic and antimicrobial peptides resistance, as well as bacterial virulence [23,24,25]. In *P. aeruginosa* the lipid A can be covalently modified through addition of 4-amino-4-deoxy-L-arabinose (L-Ara4N) by enzymes encoded by the *arnBCADTEF* operon [26], which decreases the negative charge of the lipid A and thus reduces the binding of polymyxin [27]. PhoP-PhoQ and other two-component regulatory system, PmrA-PmrB, CprR-CprS, BqsR-BqsS, and ParR-ParS directly control the expression of the *arnBCADTEF* operon [28,29,30]. In addition, PhoP-PhoQ regulates the expression of *pagP* (PA14_46900) that contributes to the palmitoylation of lipid A [31]. A previous transcriptomic analysis on a *phoP* mutant revealed that PhoP positively regulates multiple genes, including *gabD*, *gabT*, PA0921, PA1343, PA3885 and the PA4010-PA4011 and PA4456-PA4453 operons, and negatively regulates the expression of PA1196, PA3309 and PA4918 [32,33]. Electrophoretic mobility shift assay (EMSA) results demonstrated a direct regulation of PA0921 and PA1343 by PhoP [33]. Loss of function mutations in the *phoQ* gene have been identified in *P. aeruginosa* strains from cystic fibrosis receiving inhaled colistin treatment, which contribute to high level resistance to polymyxins [27]. Mutation of *phoQ* in a wild-type reference strain PAO1 reduced the bacterial twitching motility, biofilm formation, cytotoxicity as well as virulence in a lettuce leaf and a chronic rat lung infection models [34]. These results demonstrate a global regulatory role of the PhoP-PhoQ system. However, genes directly regulated by PhoP and their roles in the bacterial resistance to polymyxin B remains to be explored.

In this study, a ChIP-Seq assay was used to identify genes that are directly regulated by PhoP. We further studied the roles and functional mechanisms of those genes in the bacterial tolerance to polymyxin B. Our results revealed novel genes regulated by the PhoP-PhoQ two-component regulatory system and advanced our understanding of the polymyxin B tolerance determinants in *P. aeruginosa*.

## 2. Materials and Methods

### 2.1. Chemicals, Bacterial Strains, and Plasmids

Bacteria were cultured in Luria–Bertani (LB) broth (10 g/L tryptone, 5 g/L yeast extract and 5 g/L NaCl, pH 7.0–7.5), cation adjusted Mueller-Hinton broth (CA-MHB) (Mueller–Hinton broth dry powder (Oxoid, Basingstoke, Hampshire, England) 21 g/L, CaCl_2_ 55.5 mg/L, MgCl_2_·6H_2_O 42 mg/L, pH 7.0–7.5), second basal medium (BM2) (0.03 M glucose, 0.04 M K_2_HPO_4_, 0.022 M KH_2_PO_4_, and 0.007 M (NH_4_)_2_SO_4_, pH 7.0), M9 medium (15.12 g/L Na_2_HPO_4_·12H_2_O, 3.0 g/L KH_2_PO_4_, 0.5 g/L NaCl, 1.0g/L NH_4_Cl, 0.492 g/L MgSO_4_·7H_2_O and 3.94 g/L succinic acid, pH 7.0) or on LB agar (LB broth containing 15 g/L agar) at 37 °C aerobically. The bacterial strains and plasmids used in this study were listed in Appendix A.

### 2.2. RNA Extraction, Reverse Transcription, and Quantitative Real-Time PCR

Bacteria were grown overnight in LB at 37 °C. The culture was diluted 1:100 in fresh LB and grown to an OD_600_ of 0.8. Total RNA was isolated with a Bacteria Total RNA kit (Zomanbio, Beijing, China) and the RNA concentration was determined with a NanoDrop spectrophotometer (Thermo Scientific, Waltham, MA, USA). cDNA was synthesized by using primeScript Reverse Transcriptase (TaKaRa, Dalian, China). The cDNA was then mixed with specific primers (Appendix A) and an SYBR Premix ExTaq II (TaKaRa, Dalian, China). Quantitative real-time PCR was performed with a CFX Connect real-time system (Bio-Rad, Hercules, CA, USA). The 30S ribosomal protein gene *rpsL* was used as an internal control [35].

### 2.3. Construction of lacZ Transcriptional Fusions and β-Galactosidase Activity Measurement

Fragments of the promoter regions of the indicated genes were amplified by PCR with primers listed in Appendix A, using PA14 chromosomal DNA as the template. The PCR products were cloned into the *Sma*I-*Hin*dIII sites of the plasmid pUCP20-*lacZ* [36]. The resulting plasmid was transferred into the indicated strains. The strains were grown to an OD_600_ of 0.8. Bacterial cells from 0.5 mL culture were collected by centrifugation and resuspended with 1.5 mL Z Buffer (60 mM Na_2_HPO_4_, 60 mM NaH_2_PO_4_, 10 mM KCl, 1 mM MgSO_4_, 50 mM β-mercaptoethanol, pH 7.0; BBI LifeScience, Shanghai, China). One mL of the bacterial suspension was used to measure OD_600_. The remaining 500 μL suspension was mixed with 10 μL 0.1% SDS and 10 μL chloroform (BBI LifeScience, Shanghai, China) by vortex for 10 s. Then 100 μL O-nitrophenyl-β-D-galactopyranoside (ONPG, 40 mg/mL; Sigma, USA) was added to the mixture and incubated at 37 °C. When the color of the mixture turned yellow, the reaction was stopped by addition of 500 μL of 1.0 M Na_2_CO_3_. The reaction time was recorded. The mixture was subjected to centrifugation at 12,000× *g* for 1 min, then OD_420_ of the supernatant was measured with a spectrometer (Bio-Rad). The β-galactosidase activity (Miller units) was calculated as (1000 × OD_420_)/(T × V × OD_600_). T: Reaction time (min); V: the bacterial sample volume (mL).

### 2.4. ChIP-Seq and Data Analysis

To construct C-terminus FLAG tagged *phoP*, the *phoP* coding region was amplified by PCR using PA14 chromosomal DNA as the template with primers listed in Appendix A. The FLAG tag coding sequence was included in the primer annealing to the 3′ end of the *phoP* gene. The resulting PCR product was cloned into the *Bam*HI and *Pst*I sites of the plasmid pUCP20, resulting in pUCP20-*phoP*-FLAG.

The ChIP-Seq experiment and data analysis were performed by Wuhan IGENEBOOK Biotechnology Co., Ltd. PA14 containing the pUCP20-*phoP*-FLAG was grown in LB to an OD_600_ of 1.0. Formaldehyde was added to the culture at the concentration of 1% for 10 min at 37 °C with continuous shaking for crosslinking. A total of 125 mM glycine was added to the medium to stop the crosslinking. The bacterial cells were collected by centrifugation and washed twice a Tris buffer (20 mM Tris-HCl pH 7.5, 150 mM NaCl) containing a complete proteinase inhibitor cocktail (Roche). Then the cells were incubated in 400 μL nuclei lysis buffer (50 mM Tris-Hcl (PH8.0), 1% SDS, 1% Triton X-100, 10 mM ethylene diamine tetraacetic acid (EDTA), mini-protease inhibitor cocktail (Roche)) for 30 min. The chromosomal DNA was sonicated to the sizes of 200–500 bp (20-s with 30-s interval, 15 cycle, Diagenode Bioruptor pico), followed by centrifugation at 14,000× *g* at 4 °C for 10 min. Ten microliters (10 μL) of the supernatant was used as input. Ten micrograms (10 μg) of anti-Flag antibody (MAB 3118) was incubated with 100 μL of the supernatant for 16 h at 4 °C. 30 μL protein G magnetic beads (Life Technologies, Carlsbad, CA, USA) was added to the supernatant and incubated at 4 °C for 2 h with gentle shaking. The beads were then washed sequentially with a low salt wash buffer (20 mM Tris-HCl, 150 mM NaCl, 1% TritonX-100, 0.1% SDS, 2 mM EDTA, pH8.1), a high salt wash buffer (20 mM Tris-HCl, 500 mM NaCl, 1% TritonX-100, 0.1% SDS, 2 mM EDTA, pH8.1), a LiCl wash buffer (0.25 M LiCl, 10 mM Tris-HCl, 1% sodium deoxycholate, 1% NP40, 1 mM EDTA, pH 8.1), and TE buffer (1 mM EDTA, 10 mM Tris-HCl, pH 8). The protein-DNA complex was eluted by 400 μL elution buffer (0.1 M NaHCO_3_, 1% SDS), followed by incubation at 65 °C for 20 min. Then 20 μL of 5 M NaCl was added to 400 μL of the elute and incubated at 65 °C overnight to reverse the cross-linking, followed by incubation with 20 mg/mL Proteinase K in 30 mM Tris-HCl and 10 mM EDTA (pH 6.5) at 45 °C for 1 h. The DNA was isolated by phenol/chloroform/isoamyl extraction. The 250–350 bp DNA fragments were selected by SPRI beads. After repair and adaptor ligation, the DNA was amplified by PCR for 15 cycles. The library was sequenced with the HiSeq 2000 system (Illumina) for 50 nt single-end sequencing. Totally 38,494,074 and 50,472,658 reads were obtained from the input and ChIP samples, respectively. Low-quality reads were filtered out with Trimmomatic (v. 0.38), resulting in 32,663,770 and 43,214,422 clean reads of the input and ChIP samples, respectively. The clean reads were mapped to the PA14 genome by Bwa (v. 0.7.15). Potential PCR duplicates were removed with the software Samtools (v. 1.3.1). The software MACS2 (v. 2.1.1.20160309) was used to call peaks by default parameters (model fold, 5, 50; bandwidth, 300 bp; q value, 0.05).

### 2.5. Expression and Purification of the 6×His-Tagged PhoP Protein

The *phoP* gene was amplified by PCR with primers listed in Appendix A and cloned into the plasmid pET28a, resulting in a C-terminal 6×His-tagged *phoP* (*phoP*-His). The cloned fragment was verified by sequencing performed by GENEWIZ Corporation (Suzhou, China). The plasmid was transferred into an *E. coli* strain BL21 (DE3). The bacteria were grown in LB to an OD_600_ of 0.6, followed by induction with 0.2 mM isopropyl-β-D-thiogalactoside (IPTG) for 5 h at 37 °C with agitation at 200 rpm. The bacteria were harvested by centrifugation at 4 °C. The pellet was resuspended with a cold lysis buffer (50 mM Na_2_HPO_4_, 50 mM NaH_2_PO_4_, 0.3 M NaCl). After sonication on ice, the mixture was subjected to centrifugation at 10,000× *g* for 10 min at 4 °C. The supernatant was collected and mixed with the Ni-nitrilotriacetic acid (NTA) beads (Qiagen, Düsseldorf, North Rhine-Westphalia, GER) and incubated at 4 °C for 2 h with gentle shaking. The beads were washed three times with the lysis buffer containing 20 mM imidazole. The bound protein was eluted with the lysis buffer containing 300 mM imidazole and examined by SDS-PAGE.

### 2.6. EMSA

Sequences of the DNA probes are listed in Appendix A. DNA fragments (200 ng) were incubated with 0, 2, 4 or 8 μM purified His-tagged PhoP in a 20-μL binding reaction system [50 mM Tris, pH 7.9, 50 mM NaCl, 0.5 mM EDTA, 10% glycerol, 1% (*v*/*v*) NP-40 (Solarbio, Beijing, China)] at room temperature for 30 min [37]. The binding mixtures were loaded onto an 8% native polyacrylamide gel in 1×Tris-borate-EDTA (TBE) buffer (0.044 M Tris, 0.044 M boric acid, 0.001 M EDTA, pH 8.0) that had been pre-run on ice at 100 V for 1 h. The electrophoresis was performed on ice at 100 V for 75 min. The gel was stained with ethidium bromide in 1×TBE buffer for 5–10 min. The bands were visualized with a molecular imager ChemiDoc TM XRS+ (Bio-Rad, Hercules, CA, USA).

### 2.7. Minimum Inhibitory Concentration (MIC) Measurement

Bacteria of the indicated strains were diluted to 1 × 10^5^ CFU/mL in LB or CA-MHB, and added into each well of a 96-well plate (150 μL/well). Antibiotics with appropriate concentrations were added into the first row of the plate, followed by a serial two-fold dilution. The plate was incubated at 37 °C for 24 h. The lowest antibiotic concentration that inhibited visible growth was recorded as the MIC.

### 2.8. Bacterial Survival Assay

Overnight bacterial cultures were diluted 100-fold in fresh medium and grown to an OD_600_ of 1.0. The bacteria were diluted to 4 × 10^5^ CFU/mL in LB and mixed with polymyxin B or LL-37 at indicated concentrations. After incubation at 37 °C for 2.5 h, the number of live bacteria was determined by plating and colony counts.

### 2.9. SDS Susceptibility Assay

Overnight bacterial cultures were subcultured into a fresh LB medium to an OD_600_ of approximately 0.8. The bacteria were harvested by centrifugation and transferred to an M9 medium and grown to an OD_600_ of 1.0. The bacteria were collected by centrifugation and washed twice with the M9 medium. A total of 1 × 10^9^ CFU/mL bacteria were treated with 3.5 mM SDS in the M9 medium for 45 min at 30 °C. The number of live bacteria was determined by plating and colony counts.

### 2.10. Ethidium Bromide Influx Assay

Bacteria were grown in LB to an OD_600_ of 1.0. The bacteria were collected by centrifugation. For the polymyxin B treatment experiment, the bacteria were resuspended in fresh LB, followed by incubation with polymyxin B at the concentration of 0.039 μg/mL (0.125 MIC) or 0.078 μg/mL (0.25 MIC) for 2.5 h [38]. For the SDS treatment experiment, the bacteria were resuspended in M9 and incubated in the presence of 3.5 mM SDS for 45 min [39]. Then ethidium bromide was added to a final concentration of 2 μg/mL. After incubation at 37 °C for 15 min, the fluorescence of each sample (excitation at 530 nm, emission at 600 nm) was measured with a Luminoskan Ascent Luminometer (Varioskan Flash, Thermo Scientific, Waltham, MA, USA).

### 2.11. Dansyl-Polymyxin B Binding Assay

The dansyl-polymyxin B was synthesized as previously described [40]. Briefly, 40 mg of polymyxin B sulfate and 0.42 g NaHCO_3_ were dissolved in 50 mL ddH_2_O. Ten milligrams (10 mg) of dansyl chloride was dissolved in 0.8 mL acetone. Then the two solutions were mixed and incubated at room temperature for 90 min in dark. A total of 12.5 g of G-25 Sephadex bead was added to a 2.5 cm diameter purification column, and treated with at least 100 mL balance buffer (NaH_2_PO_4_ 1.4196 g, NaCl 8.4738 g, ddH_2_O 1 L, pH 7.10). The mixture was loaded into the column, eluted with the balance buffer. The eluent that appeared bright yellow under ultraviolet light was the dansyl-polymyxin B. The collected dansyl-polymyxin B was then extracted with 1/2 volume of butanol and dried by evaporation. The dansyl-polymyxin B powder was dissolved in N-2-hydroxyethylpiperazine-N-2-ethane sulfonic acid (HEPES), pH 7.0, and stored at −20 °C.

Bacteria of the indicated strains were grown in LB to an OD_600_ of 1.0. The bacterial cells were collected by centrifugation and washed twice with normal saline (0.9% NaCl). The bacterial cells were resuspended in normal saline to reach 1 × 10^9^ CFU/mL and incubated with or without 0.26 μg/mL dansyl-polymyxin B for 5 min at 30 °C in dark. Then the bacteria were washed twice with normal saline and resuspended in 1 mL normal saline. A total of 150 μL of the suspension was transferred into each well of a black 96-well microtiter plate (Nunc). The fluorescence (excitation at 340 nm, emission at 485 nm) was measured with a Luminoskan Ascent Luminometer (Varioskan Flash, Thermo Scientific, Waltham, MA, USA). The relative fluorescence intensity alter rate of dansyl-polymyxin B was calculated as (100 × (F-F_0_)/F_0_)%. F: Fluorescence intensity of samples with dansyl-polymyxin B; F_0_: Fluorescence intensity of samples without dansyl-polymyxin B.

### 2.12. Outer Membrane Permeability Assay

The outer membrane permeability assay was performed as previously described [41,42]. Overnight cultures of indicated strains were transferred to fresh LB and grown to an OD_600_ of 1.0. Bacterial cells were washed twice with 5 mM HEPES containing 5 mM glucose (pH 7.0) and resuspended in the HEPES buffer to an OD_600_ of 0.5, followed by incubation with 10 μM NPN at 25 °C for 30 min. Then the bacteria were incubated with or without 0.78 μg/mL polymyxin B at 37 °C for another 30 min. The fluorescence intensities (excitation wavelength: 350 nm, emission wavelength: 420 nm) were measured with a Luminoskan Ascent Luminometer (Varioskan Flash, Thermo Scientific, Waltham, MA, USA). The relative fluorescence intensity was calculated as (100(F – F_0_) / F_0_)%. F and F_0_ represent fluorescence intensities of samples with polymyxin B and without polymyxin B treatment, respectively.

### 2.13. Inner Membrane Integrity Assay

The inner membrane integrity assay was performed as previously described [43,44]. Briefly, bacteria at an OD_600_ of 1.0 were washed twice with phosphate buffer saline (PBS, pH 7.2) and resuspended to an OD_600_ of 0.5. The bacteria were incubated with 10 μM PI at 25 °C for 30 min. Then the bacterial samples were incubated with or without 0.78 μg/mL polymyxin B at 37 °C for 1 h. The fluorescence values of the samples were detected under the conditions of excitation wavelength 535 nm and emission wavelength 615 nm with a Luminoskan Ascent luminometer (Varioskan Flash, Thermo Scientific, Waltham, MA, USA). The result of relative fluorescence intensity was calculated as (100(F–F_0_) / F_0_)%. F and F_0_ represent fluorescence intensities of samples with polymyxin B and without polymyxin B treatment, respectively.

### 2.14. Statistical Analysis

All experiments were carried out at least in triplicate. Statistical significance was evaluated by the Prism software (Graphpad Software) via a one-way analysis of variance (ANOVA).

## 3. Results

### 3.1. Identification of Genes Directly Regulated by PhoP in P. aeruginosa

To identify genes directly regulated by the PhoP, we overexpressed a FLAG tagged PhoP in the wild-type PA14 and performed ChIP-Seq. The DNA binding loci and enrichment folds were listed in Appendix A. The intergenic regions that were enriched more than 2-fold are shown in Table 1. Consistent with previous reports, the promoter regions of *oprH*, PA14_46900 and *arnB* were enriched by 17.2-, 15-, and 6.9-fold, respectively. In a previous study, the PhoP binding sequence was predicted as CGTTCAGNNNNNRTTCAG [32]. A multiple expression motifs for elicitation (MEME) analysis of our ChIP-Seq peak regions revealed the potential PhoP binding motif as G/ATTCAG (Figure 1A), which is similar to the repeated sequence in the previously predicted PhoP consensus binding sequence (underlined). The potential PhoP binding motif was found in the regions upstream of the open reading frames of PA14_46900, PA14_50740, PA14_50750, PA14_52340, and the operons of PA14_11970-PA14_11960 (within the open reading frame of PA14_11980) and PA14_52350-PA14_52370, as well as the regions between the 3′ ends of the coding regions of PA14_21870 and PA14_21860 (Figure 1B). Although the promoter region of *pilY1* was enriched in the ChIP-Seq assay, no G/ATTCAG-like sequence was found upstream of the *pilY1* coding region (Figure 1B).

To verify whether PhoP-PhoQ controls the expression of those genes, we performed quantitative real time PCR assays. Mutation of *phoP* and *phoQ* individually or simultaneously did not affect the expression level of *pilY1*, PA14_11980, PA14_21860 or PA14_21870 (Figure 1C). However, deletion of the *phoQ* gene increased the expression levels of PA14_11970, PA14_46900, PA14_50740, PA14_52340 and PA14_52350 (Figure 1C). Meanwhile, deletion of *phoP* or both *phoP* and *phoQ* reduced the expression of these genes (Figure 1C). In wild-type PA14, the expression of PA14_11970, PA14_46900, PA14_50740, PA14_52340, and PA14_52350 was induced by a low Mg^2+^ growth condition compared to the high Mg^2+^ condition [45], which was abolished by the deletion of *phoP* (Figure 1D).

To confirm the promoters of these genes are regulated by PhoP, we constructed transcriptional fusions between each of the promoters and a *lacZ* gene, resulting in P_PA14_11970_-*lacZ*, P_PA14_46900_-*lacZ*, P_PA14_50740_-*lacZ*, P_PA14_52340_-*lacZ* and P_PA14_52350_-*lacZ*. The LacZ activities were decreased in the Δ*phoP* and Δ*phoP*Δ*phoQ* mutants, but increased in the Δ*phoQ* mutant (Figure 2A–E). We then performed EMSA to verify the direct binding between PhoP and these promoters. Consistent with a previous report [32], the purified PhoP protein bond to the promoter region of PA14_46900 (Figure 2F). In addition, band shifts were observed with the promoter regions of, PA14_50740, PA14_11970-PA14_11980, and PA14_52340-PA14_52350 (Figure 2F). In combination, these results suggested that the PhoP-PhoQ two component regulatory system directly controls the expression of PA14_46900, PA14_50740 as well as the operons of PA14_52350-PA14_52370 and PA14_11970-PA14_11960 in response to Mg^2+^ concentrations. Based on the conserved the domains and previous studies, the functions and the designated names of the PhoP-PhoQ regulated genes were listed in Table 2.

Previous studies demonstrated that, besides PhoP-PhoQ, the two-component regulatory systems PmrA-PmrB, BqsS-BqsR, ParS-ParR, and CprS-CprR are also involved in the regulation of the *arnBCADTEF* operon [29,30,51,52]. We thus examined whether the four two-component regulatory systems control the expression of the genes listed in Table 2. Mutants of the individual genes were picked up from the PA14 transposon mutant library [53]. As shown in Appendix A, the expression levels of *papP*, *mpl*, *pagP*, *slyB*, PA14_52340, *ppgS*, and *ppgH* were similar between the mutants and the wild-type PA14, indicating that the four regulatory systems might not control the expression of these genes.

### 3.2. Roles of PhoP-PhoQ Regulated Genes in the Bacterial Tolerance to Polymyxin B

A previous study revealed that *pagP* encodes for a lipid A palmitoyltransferase which transfers palmitate to LPS [31]. However, its role in the bacterial tolerance to polymyxin B was not known. *ppgS* and *ppgH*, which encodes an alanyl-phosphatidylglycerol synthase and hydrolase, respectively, had been shown to contribute to lipid homeostasis in cell membrane [50]. The functions of the other genes as well as the genes in the same operons, namely *slyB*, PA14_52340, and *mpl*-*papP* remain to be studied.

To examine the roles of the PhoP-PhoQ regulated genes in the bacterial resistance to polymyxin B, we selected strains with mutations in each of the genes from the PA14 transposon insertion library [53]. The mutants displayed the same MIC of polymyxin B as the wild-type PA14 (Table 3). We then examined their roles in the bacterial tolerance to polymyxin B. The strains of *papP*::Tn, *mpl*::Tn, *pagP*::Tn, *slyB*::Tn, *ppgS*::Tn, and *ppgH*::Tn displayed lower survival rates than that of the wild-type PA14, whereas the strains of PA14_11980::Tn and PA14_52340::Tn displayed similar survival rates as the wild-type strain (Figure 3A).

To further confirm the roles of these genes in the bacterial tolerance to polymyxin B, we constructed deletion mutants in wild-type PA14. Deletion of *papP*, *mpl*, *pagP*, *slyB*, *ppgS*, and *ppgH* reduced the bacterial survival rates after polymyxin B treatment. Complementation with the corresponding genes restored the survival rates (Figure 3B). Consistent with the results of survival rate, treatment with polymyxin B resulted in higher ethidium bromide (EtBr) influx in the mutants, indicating more severe membrane damage (Figure 3C). The human cationic antimicrobial peptide LL-37 kills bacteria via binding to LPS and subsequent damage of the cell membrane, which is similar to polymyxins [54]. Deletion of *papP*, *mpl*, *pagP*, *slyB*, *ppgS*, and *ppgH* also increased bacterial susceptibility to LL-37 (Figure 3D).

### 3.3. Mechanisms of Polymyxin B Tolerance Mediated by the Identified Genes

One of the major mechanisms of the bacterial tolerance to polymyxin B is to reduce the binding between lipid A and polymyxin B [55]. We thus used a dansyl chloride labeled polymyxin B to measure the amount of surface associated polymyxin B [55]. Previous studies revealed a role of PagP in the palmitoylation of lipid A [56]. In agreement with its function, deletion of *pagP* increased the amount of the surface associated polymyxin B (Figure 4A). However, mutation in the other genes did not affect the binding of polymyxin B to the cell (Figure 4A).

We then examined the roles of the genes in response to membrane damage. Treatment with SDS resulted in more cell death and EtBr influx in the strains of Δ*papP*, Δ*mpl*, Δ*slyB*, Δ*ppgS* and Δ*ppgH* (Figure 4B,C). Simultaneous deletion of the five genes (designated as Δ5) further decreased the bacterial survival rate upon SDS treatment (Figure 4D). However, deletion of *pagP* in wild-type PA14 and the Δ5 mutant (designated as Δ6) did not reduce the bacterial resistance to SDS compared to the corresponding parental strains (Figure 4D), indicating that *papP*, *mpl*, *slyB*, *ppgS*, and *ppgH* are involved in bacterial response to membrane damage.

We next dissected the roles of the five genes in maintaining outer and inner membrane integrity by NPN and PI staining, respectively. After treatment with polymyxin B, the Δ*phoP* and Δ*phoQ* mutants displayed higher and lower NPN/PI stains than the wild-type PA14, respectively (Figure 5A,B), demonstrating an important role of the PhoP-PhoQ regulatory system in protecting the membranes against polymyxin B. Deletion of *papP* or *slyB* increased the NPN staining whereas deletion of *mpl*, *ppgS* and *ppgH* did not affect the NPN staining (Figure 5A). SlyB is predicted to be a lipoprotein that localized on outer membrane [57]. It might inhibit the insertion of polymyxin B into the outer membrane or play a role in membrane repair [49]. PapP is predicted to localize in the inner membrane and contains a potential type 2 phosphatidic acid phosphatase domain [57,58]. It might be involved in the modification of outer membrane which contributes to bacterial tolerance to polymyxin B.

Meanwhile, deletion of each of the five genes increased PI staining after polymyxin B treatment (Figure 5B). Considering the roles of PapP and SlyB in reducing the outer membrane damage, the increased PI staining might be due to higher amount of polymyxin B that crossed the outer membrane. Since mutation of *mpl*, *ppgS*, or *ppgH* did not affect the polymyxin B triggered outer membrane damage, the higher PI staining indicates that these genes might be involved in maintaining the inner membrane integrity. Mpl contains a potential methylpurine-DNA glycosylase domain and is predicted to localize in cytoplasm [57,58]. However, its physiological role remains unknown. PpgS and PpgH are inner membrane proteins involved in the modification of the phosphatidylglycerol in cell membrane, which might reduce the insertion of polymyxin B into the inner membrane.

We then determined the cumulative effect of the PhoP regulated genes in bacterial resistance to polymyxin B. The Δ5 mutant displayed the same MIC of polymyxin B as the wild-type strain, whereas the Δ6 and the Δ*phoP* mutants displayed lower MICs in LB (Table 3) or CA-MHB medium (Appendix A). The survival rates of the Δ5 and Δ6 mutants were lower than that of the wild-type strain following polymyxin B treatment (Figure 5C). Compared to the Δ*phoP* mutant, the Δ6 mutant displayed a 4.7-fold lower survival rate (Figure 5C). In combination, these results indicate that *papP*, *mpl*, *slyB*, *ppgS*, and *ppgH* contribute to bacterial resistance to polymyxin B by maintaining membrane integrity while *pagP* reduces the binding of polymyxin B to the bacterial LPS.

## 4. Discussion

In this study, we identified novel PhoP regulated genes by ChIP-Seq, including *papP* (PA14_11960), *mpl* (PA14_11970), *slyB* (PA14_50740), *ppgS* (PA14_52350), and *ppgH* (PA14_52370), and demonstrated that these genes contribute to the bacterial tolerance to polymyxin B.

A previous study revealed that mutation of *phoQ* attenuates the bacterial virulence in a murine bacteremia infection model which is due to the upregulation of *oprH* that serves as a binding target of the complement component C3 [59]. In addition, mutation of *phoQ* reduces the bacterial twitching motility, biofilm formation, cytotoxicity as well as virulence in a lettuce leaf and a chronic rat lung infection model [34]. Microarray analyses demonstrated that mutation of *phoQ* altered the expression of 474 genes [34]. In addition to the *arnBCADTEF* and *pmrAB* operons, mutation of *phoQ* resulted in upregulation of genes involved in alginate, *Pseudomonas* quinolone signal (PQS) and pyoverdine synthesis as well as *slyB* [34].

*pagP* encodes a palmitoyltransferase that is regulated by PhoP-PhoQ and contributes to the palmitoylation of lipid A and the bacterial resistance to a synthetic cationic antimicrobial peptide C18G [31]. However, mutation of *pagP* in a Δ*phoQ* mutant did not affect the bacterial resistance to polymyxin B in colistin agar dilution and polymyxin B plate assays [31]. We suspect that the overexpression of the *arnBCADTEF* operon in the Δ*phoQ* mutant lead to a high-level L-Ara4N addition to lipid A, which might compensate for the loss of the palmitoylation. Here we found that mutation of *pagP* in wild-type PA14 decreased the bacterial survival rate in a polymyxin killing assay in liquid LB. However, simultaneous mutation of *pagP* and *arnB* resulted in a similar survival rate as the *pagP* or *arnB* mutant (Appendix A), indicating a possible redundancy of the modifications. Further studies are required to examine whether the palmitoylation of lipid A interfere with the addition of Ara4N, and vice versa.

SlyB is an outer membrane lipoprotein that has been shown to be involved in the stabilization of the outer membrane in *E. coli* and *S. enterica* [60]. A study in *Burkholderia multivorans* demonstrated that mutation of *slyB* impairs the bacterial growth in presence of EDTA, SDS or the iron (III) chelator ethylenediaminedi (o-hydroxyphenylacetic) acid (EDDHA) [49]. In *P. aeruginosa*, the expression of *slyB* has been shown to be regulated by AlgU, an alternative sigma factor that response to periplasmic stresses [61,62]. Here we found that SlyB contributes to the bacterial survival under polymyxin B and SDS treatment. The NPN staining assay revealed a role of SlyB in maintaining outer membrane integrity in response to polymyxin B. These results indicate a conserved role of SlyB in maintaining outer membrane integrity. However, the functional mechanism of SlyB remains to be elucidated.

PpgS and PpgH are localized in the cell membrane and function as alanyl-phosphatidylglycerol synthase and alanyl-phosphatidylglycerol hydrolase, respectively [63,64,65]. The two enzymes are involved in the homeostasis of aminoacylation of the negatively charged headgroup of phosphatidylglycerol (PG), which affects the membrane surface charge, morphology, fluidity and contributes to maintaining the integrity of the cytoplasm membrane in response to environmental stresses [63,65]. Mutation of *ppgH* reduced the bacterial resistance to Cr^3+^, the cationic peptide protamine, the β-lactam antibiotic cefsulodin and an osmolality stress (625 mM sodium lactate) [63], and mutation of *ppgS* reduced the bacterial resistance to a variety of antibiotics including ampicillin, cefsulodin, daptomycin, and polymyxins [65]. Through the NPN and PI staining assays, we indeed found that PpgS and PpgH contribute to inner, but not outer, membrane integrity following polymyxin B treatment, indicating an important role of the homeostasis of the aminoacylation of PG in the bacterial resistance to polymyxin B.

PapP and mpl contain a phosphatidic acid phosphatase type 2/haloperoxidase and a methylpurine-DNA glycosylase domain, respectively [58]. However, further studies are needed to understand how the two proteins contribute to membrane integrity in response to polymyxin B. The PhoP-PhoQ mediated regulatory pathways and bacterial polymyxin resistance mechanisms was summarized in Figure 6.

In combination, our results reveal that the PhoP-PhoQ two-component regulatory system contributes to bacterial tolerance to polymyxin B by directly regulating genes involved in LPS modification and membrane integrity maintenance. These genes may also be involved in the bacterial response to environmental stresses such as cation depletion and host antimicrobial substances, which deserves further investigation.

## Figures and Tables

**Figure 1 microorganisms-09-00344-f001:**
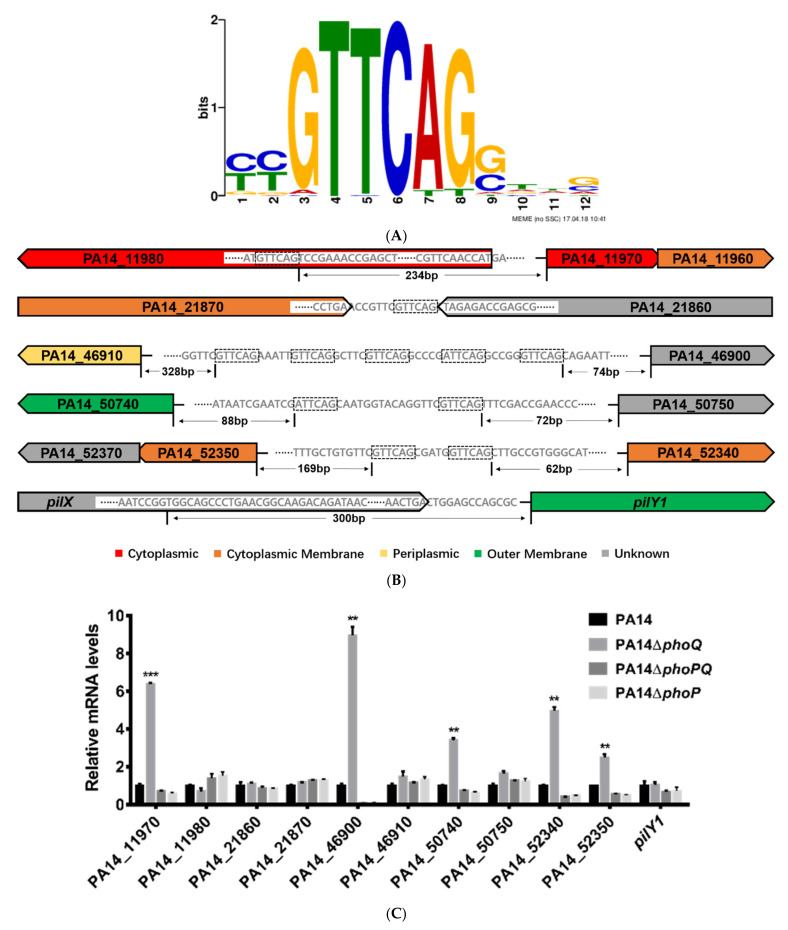
Genes controlled by the PhoP-PhoQ two-component regulatory system. (**A**) Potential PhoP binding motif was identified by MEME from the ChIP-seq peak regions. (**B**) Locations of the enriched peaks on the PA14 chromosome. The predicted *phoP* binding sequences are boxed. Colors of the genes were adopted from the Pseudomonas Genome Database (www.pseudomonas.com). (**C**) The wild-type PA14, Δ*phoP*, Δ*phoQ*, and Δ*phoP*Δ*phoQ* mutants were grown to an OD_600_ of 1.0. The mRNA levels of the indicated genes were determined by quantitative real time PCR. (**D**) The wild-type PA14, Δ*phoP*, and Δ*phoQ* mutants were grown to an OD_600_ of 1.0 in the BM2 medium with high Mg^2+^ (2 mM) and low Mg^2+^ (2 μM). The mRNA levels of the indicated genes were determined by quantitative real time PCR. Data represents mean ± standard deviation. *, *p* < 0.05; **, *p* < 0.01; ***, *p* < 0.001 by Student’s *t*-test.

**Figure 2 microorganisms-09-00344-f002:**
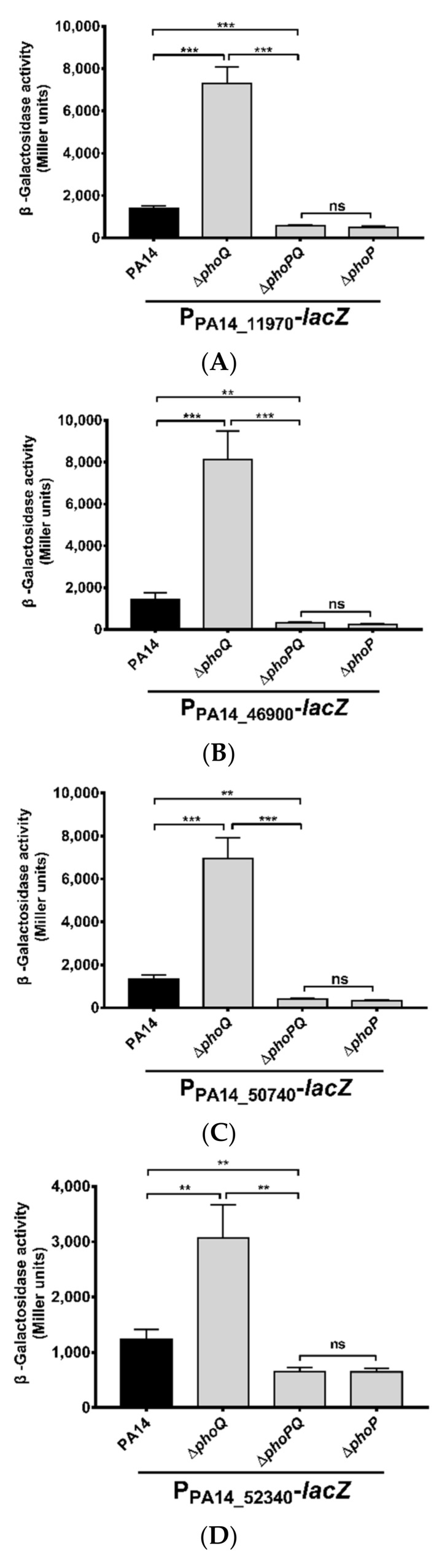
Promoters regulated by the PhoP-PhoQ two-component regulatory system. The transcriptional fusions of P_PA14_11970_-*lacZ* (**A**), P_PA14_46900_-*lacZ* (**B**), P_PA14_50740_-*lacZ* (**C**), P_PA14_52340_-*lacZ* (**D**), and P_PA14_52350_-*lacZ* (**E**) were transferred into PA14 and the *phoP*, *phoQ*, and *phoP-phoQ* mutants. The bacteria were cultured in LB at 37 °C to an OD_600_ of 1.0. The values (Miller units) are the means of three experiments. ns, not significant. **, *p* < 0.01; ***, *p* < 0.001, by Student’s *t*-test. (**F**) Interactions between PhoP and its target DNAs were examined by EMSA. Purified 6×His-tagged PhoP was incubated with the promoter regions of the indicated genes (see Appendix A). The internal fragment within the *phoP* coding region was used as a negative control. Arrows indicate the positions of unbound probe and PhoP-probe complex.

**Figure 3 microorganisms-09-00344-f003:**
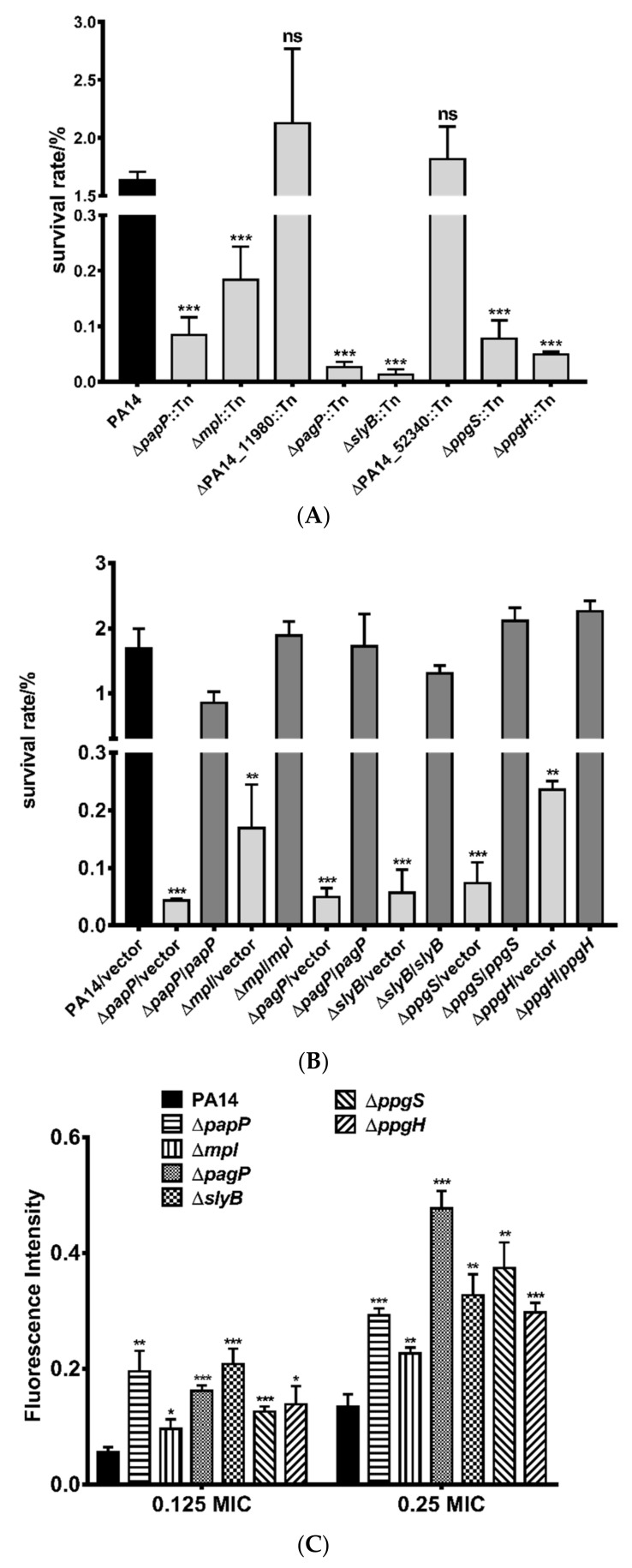
Roles of the PhoP regulated genes in the bacterial tolerance to polymyxin B and LL-37. Survival rates of the PA14 transposon insertion mutants (**A**), corresponding in-frame deletion mutants and respective complemented strains (**B**) after polymyxin B treatment. The strains were treated with polymyxin B (0. 78 μg/mL) for 2.5 h at 37 °C. Error bars represent standard errors. ns, not significant. **, *p* < 0.01, ***, *p* < 0.001, compared to wild-type PA14 by Student’s *t* test. (**C**) The indicated strains were grown in LB to an OD_600_ of 1.0 and treated with 0.039 μg/mL (0.125 MIC) or 0.078 μg/mL (0.25 MIC) polymyxin B for 2.5 h. Then the bacterial cells were stained with 2 μg/mL ethidium bromide, followed by fluorescence intensity determination. *, *p* < 0.05; **, *p* < 0.01; ***, *p* < 0.001, compared to PA14 by Student’s *t*-test. (**D**) The indicated strains were grown in LB to an OD_600_ of 1.0 and treated with 200 μg/mL LL-37 for 2.5 h at 37 °C. The bacterial survival rates were determined by serial dilution and plating. **, *p* < 0.01, compared to the wild-type PA14 by Student’s *t*-test.

**Figure 4 microorganisms-09-00344-f004:**
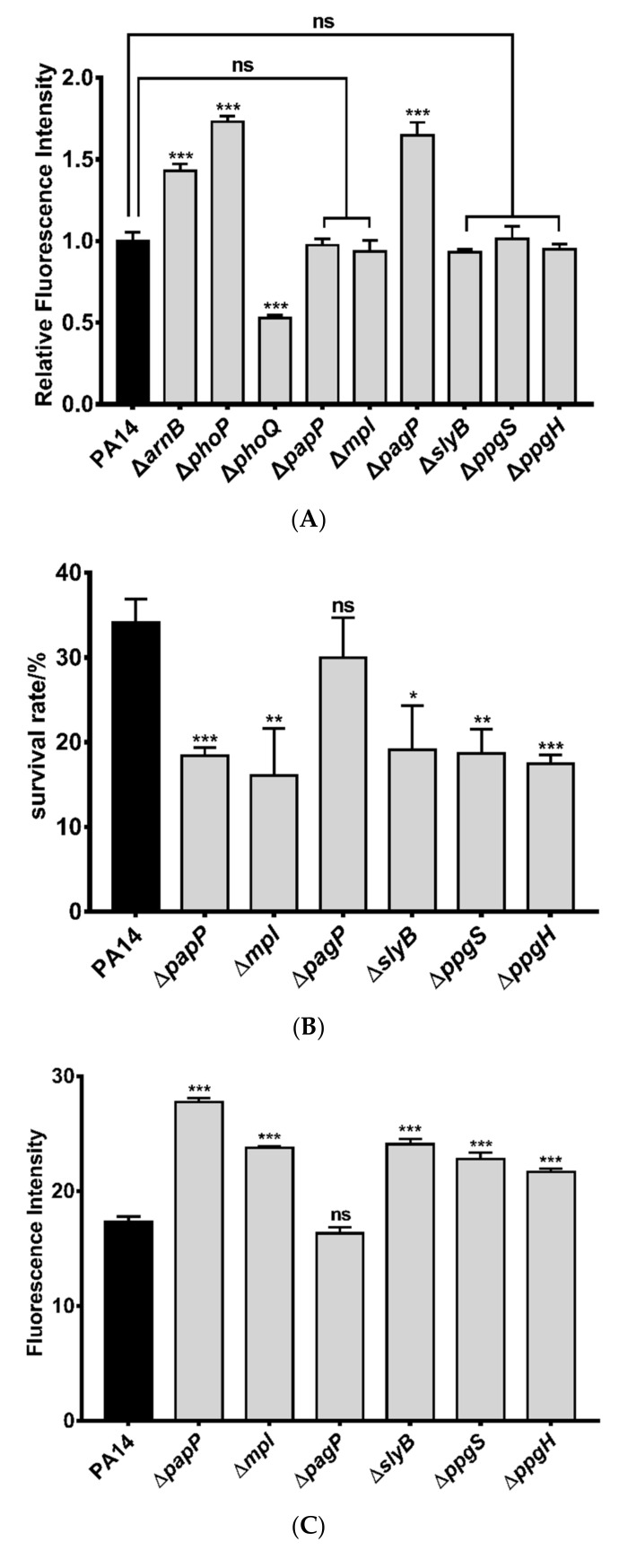
Roles of PhoP regulated genes in the bacterial tolerance to polymyxin B and SDS. (**A**) Dansyl-polymyxin B binding assay. Wild-type PA14 and the indicated mutants were treated with dansyl-polymyxin B (0.26 μg/mL) for 5 min in saline at 30 °C in dark. The bacteria were washed twice with saline and the fluorescence intensities were determined with a luminometer. The data shown represents the results from three independent experiments. ns, not significant; ***, *p* < 0.001 compared to the wild-type PA14 by Student’s *t*-test. (**B**,**C**) The indicated strains were grown to an OD_600_ of 1.0 in the M9 medium, followed by treatment with 3.5 mM SDS at 37 °C for 45 min. The bacteria were washed once with M9. Then the bacteria were subjected to serial dilution and plating on LB agar for CFU enumeration (**B**) or staining with 2 μg/mL ethidium bromide and fluorescence intensity measurement (**C**). Data shown represent results from three independent experiments. ns, not significant; *, *p* < 0.05; **, *p* < 0.01; ***, *p* < 0.001, compared to the wild-type PA14 by Student’s *t*-test. (**D**) Wild-type PA14 and the indicated mutants were grown in the M9 medium to an OD_600_ of 1.0. The bacteria were treated with 3.5 mM SDS for 45 min. The live bacteria numbers were determined by serial dilution and plating. ns, not significant; **, *p* < 0.01 by Student’s *t*-test.

**Figure 5 microorganisms-09-00344-f005:**
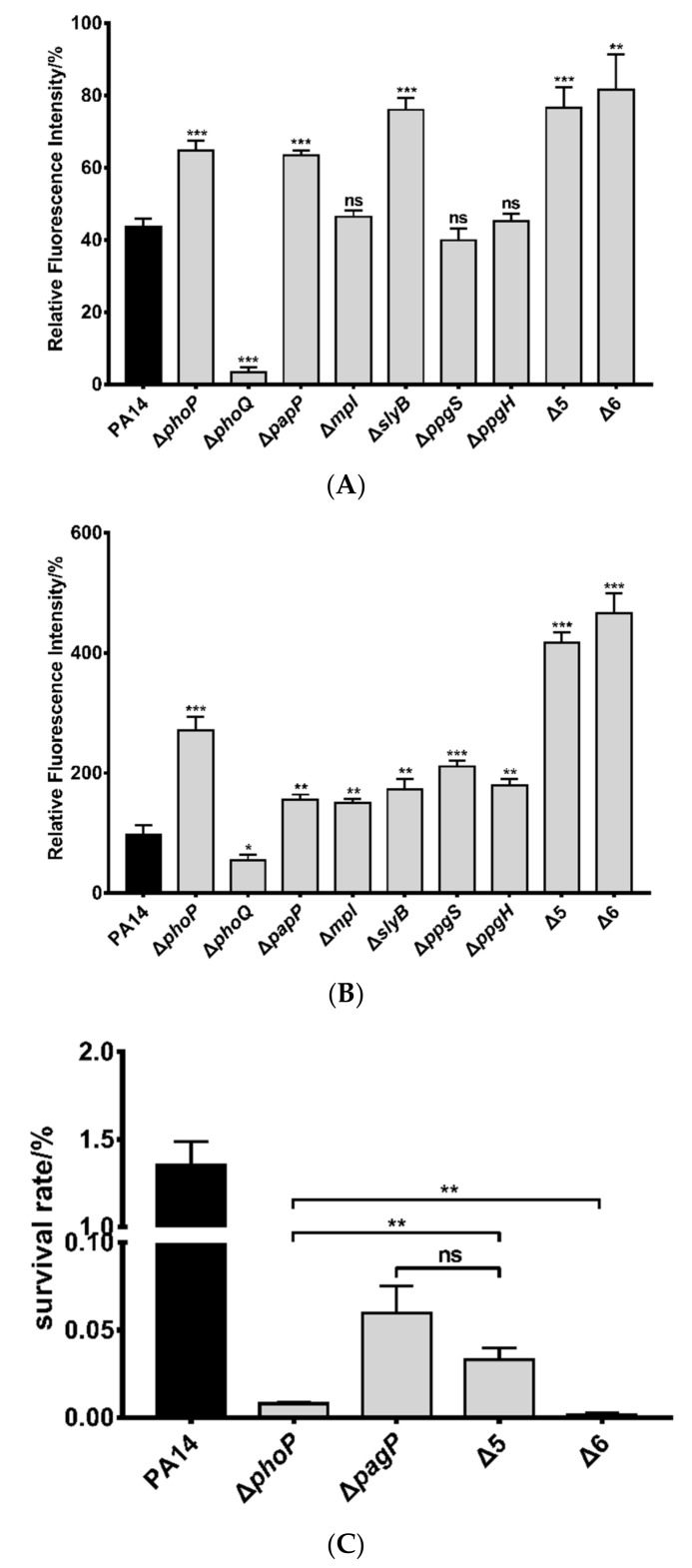
Roles of PhoP regulated genes in maintaining outer and inner membrane integrity. (**A**) Bacteria at an OD_600_ of 1.0 were collected and washed twice with 5 mM HEPES containing 5 mM glucose. The bacteria were resuspended in the HEPES buffer to an OD_600_ of 0.5, followed by incubation with 10 μM NPN at 25 °C for 30 min. Then the bacteria were incubated with or without 0.78 μg/mL polymyxin B at 37 °C for 30 min, followed by fluorescence measurement. (**B**) Bacteria at an OD_600_ of 1.0 were washed twice with PBS (pH 7.2) and resuspended in PBS to an OD_600_ of 0.5. The bacteria were incubated with 10 μM PI at 25 °C for 30 min. Then the bacterial samples were incubated with or without 0.78 μg/mL polymyxin B at 37 °C for 1 h, followed by fluorescence measurement. ns, not significant. *, *p* < 0.05; **, *p* < 0.01; ***, *p* < 0.001 compared to PA14 by Student’s *t*-test (**C**) Wild-type PA14 and the indicated mutants were grown to an OD_600_ of 1.0. The bacteria were treated with 0.78 μg/mL polymyxin B for 2.5 h at 37 °C. The live bacteria numbers were determined by serial dilution and plating. ns, not significant; **, *p* < 0.01, by Student’s *t*-test.

**Figure 6 microorganisms-09-00344-f006:**
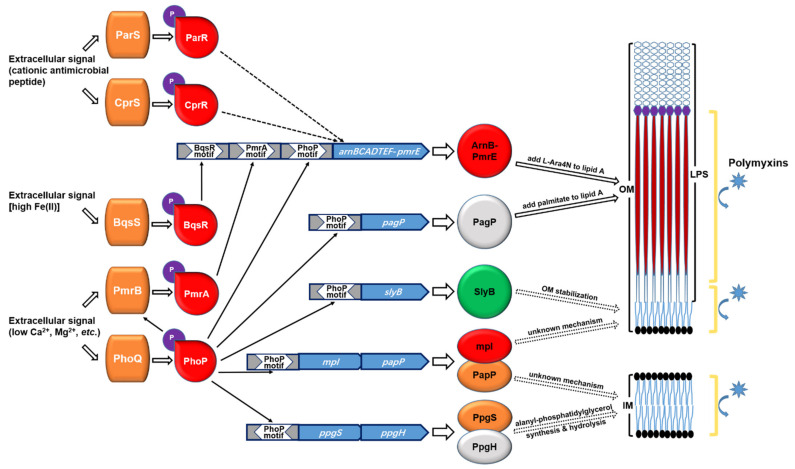
Schematic diagrams of PhoP-PhoQ regulated genes in the bacterial resistance to polymyxin B. The two-component regulatory systems PhoP-PhoQ and PmrA-PmrB are activated by depletion of divalent cations. Phosphorylated PmrA activates the transcription of *arnBCADTEF-pmrE* operon that contributes to bacterial resistance to polymyxins by adding L-Ara4N to lipid A (26). The two-component regulatory systems ParS-ParR and CprS-CprR activate expression of the *arnBCADTEF-pmrE* operon in response to cationic antimicrobial peptides [29]. Another two-component regulatory system BqsS-BqsR activates the expression of the *arnBCADTEF-pmrE* operon in response to Fe(II) [30]. Phosphorylated PhoP activates the expression of *pmrA-pmrB* and the *arnBCADTEF-pmrE* operon as well as *pagP*, *mpl*-*papP*, *slyB*, and *ppgS*-*ppgH*. The *pagP* reduces the affinity between LPS and polymyxins through addition of palmitate to lipid A. The *papP* and *slyB* protect bacterial outer membrane integrity. The *mpl* and *ppgS*-*ppgH* are involved in maintaining inner membrane integrity in response to polymyxins. Colors of the genes were adopted from the Pseudomonas Genome Database (www.pseudomonas.com).

**Table 1 microorganisms-09-00344-t001:** Potential PhoP regulated genes identified via ChIP-seq.

Genes	Summits in PA14 Chromosome	Fold Enrichment
*oprH*	4372078	17.23
PA14_46900, PA14_46910	4177576	15.03
*arnB*	1578361	6.85
PA14_50740, PA14_50750	4508762	6.24
PA14_21860, PA14_21870	1900312	5.29
PA14_52340, PA14_52350	4644776	4.87
PA14_11970, PA14_11980	1035748	4.17
*pilY1*	5372861	2.85

**Table 2 microorganisms-09-00344-t002:** Functions of the identified PhoP regulated genes.

PA14 Locus Tag	PAO1 Locus Tag	Functional Description	Product Name	Preliminary Phenotypic Analysis
PA14_11960	PA4011	PAP2 superfamily protein/DedA family protein	*papP ** (phosphatidic acid phosphatase in *P.a*)	① Upregulated by low concentrations of Mg^2+^ [32] ② Peptidoglycan recycling [46]
PA14_11970	PA4010	3-methyladenine DNA glycosylase	*mpl **	
PA14_46900	PA1343	palmitoyltransferase	*pagP*	① Directly regulated by PhoP-PhoQ [32] ② Transfers palmitate to lipid A [31]
PA14_50740	PA1053	outer membrane lipoprotein	*slyB*	① Downregulated by high concentrations of Mg^2+^ via the PhoP/PhoQ two-component system, and participate in the stabilization of the outer membrane (*E.coli* & *S.enterica*) [47,48] ② Contributes to the integrity of cell envelope (*B. multivorans*) [49]
PA14_52340	PA0921	hypothetical protein	*/*	Directly regulated by PhoP-PhoQ [32]
PA14_52350	PA0920	alanyl-phosphatidylglycerol synthase	*ppgS ** (phosphatidylglycerol synthases)	Involved in polymyxin E MIC and might altered bacterial membrane potential (a conjecture) [50]
PA14_52370	PA0919	alanyl-phosphatidylglycerol hydrolase	*ppgH ** (phosphatidylglycerol hydrolase)	

*, gene names designated in this study.

**Table 3 microorganisms-09-00344-t003:** Bacterial susceptibilities to polymyxin B.

Strain	MIC (μg/mL)
PA14	0.3125
Δ*phoP*	0.1563
Δ*papP*::Tn	0.3125
Δ*mpl*::Tn	0.3125
ΔPA14_11980::Tn	0.3125
Δ*pagP*::Tn	0.3125
Δ*slyB*::Tn	0.3125
ΔPA14_52340::Tn	0.3125
Δ*ppgS*::Tn	0.3125
Δ*ppgH*::Tn	0.3125
Δ*papP*	0.3125
Δ*mpl*	0.3125
Δ*pagP*	0.3125
Δ*slyB*	0.3125
Δ*ppgS*	0.3125
Δ*ppgH*	0.3125
Δ5 ^a^	0.3125
Δ6 ^b^	0.1563

^a^ Δ5, Δ*papP*Δ*mpl*Δ*slyB*Δ*ppgS*Δ*ppgH*; ^b^ Δ6, deletion of *pagP* in Δ5.

## Data Availability

The data presented in this study are available on request from the corresponding author.

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
