# Peer review of "Identification of Novel phoP-phoQ Regulated Genes that Contribute to Polymyxin B Tolerance in Pseudomonas aeruginosa"

_microorganisms, 2021, doi:10.3390/microorganisms9020344_

Round 1

Reviewer 1 Report

In this paper, Yang and colleagues report novel PhoP-PhoQ regulated genes of P. aeruginosa identified through their ChIP-Seq assay. Deletion of some of these genes increased susceptivity toward antimicrobial peptide Polymyxin B. Moreover, the authors dissect the cellular compartment where respective gene deletions impair, through 1-N-phenylnaphthyl-amide and propidium iodide staining assays. These findings may contribute us to develop new treatment technique against infectious diseases of gram-negative bacteria.

As listed below, there are several incompleteness of data and description of the experimental procedures, which should be corrected before acceptance of this manuscript for publication.

Line 73. At the first appearance in the manuscript, explain what the abbreviation “EMSA” means.

Line 90. There should be a description of M9 medium.

Line 120. Provide information about immunoprecipitation, including how FLAG-tagged PhoP was expressed, how DNA fragments were bound and precipitated.

Line 430. Fig 6 should read Fig 5.

Fig 3. Panels B and C are labeled contrariwise.

Fig 3 and Fig 4. The vertical axis label of panel C should not be relative OD600 but fluorescence intensity.

Table S2. Provide information about plasmd encoding FLAG tagged PhoP.

Author Response

In this paper, Yang and colleagues report novel PhoP-PhoQ regulated genes of P. aeruginosa identified through their ChIP-Seq assay. Deletion of some of these genes increased susceptivity toward antimicrobial peptide Polymyxin B. Moreover, the authors dissect the cellular compartment where respective gene deletions impair, through 1-N-phenylnaphthyl-amide and propidium iodide staining assays. These findings may contribute us to develop new treatment technique against infectious diseases of gram-negative bacteria.

As listed below, there are several incompleteness of data and description of the experimental procedures, which should be corrected before acceptance of this manuscript for publication.

Line 73. At the first appearance in the manuscript, explain what the abbreviation “EMSA” means.

Response: We added the explanation of “EMSA”.

Line 90. There should be a description of M9 medium.

Response: We added description of M9 medium in line 92.

Line 120. Provide information about immunoprecipitation, including how FLAG-tagged PhoP was expressed, how DNA fragments were bound and precipitated.

Response: Information of the immunoprecipitation has been added to the Method section.

Line 430. Fig 6 should read Fig 5.

Response: We are sorry for the mistake. The number has been corrected.

Fig 3. Panels B and C are labeled contrariwise.

Response: We are sorry for the mistake. The labeling has been corrected.

Fig 3 and Fig 4. The vertical axis label of panel C should not be relative OD600 but fluorescence intensity.

Response: The labeling has been corrected.

Table S2. Provide information about plasmid encoding FLAG tagged PhoP.

Response: The information has been added to Table S2. Information of the primers for the construction of the FLAG tagged phoP has been added to Table S3.

Reviewer 2 Report

In this article the authors examine in detail the activity of the PhoP- PhoQ two-component regulatory system on the activity of polymyxin B and polymyxin E (colistin) against Pseudomona aeruginosa known as a very resistant microorganism for which the above antibiotics constitute the last resort. The aforementioned system has the capacity of maintaining the integrity of the cellular membrane so that the bacterial linking between the polymyxins B and E positively charged and the negatively charged phosphate groups of lipopolysaccharide (LPS) on the bacterial surface of Pseudomonas is prevented. The aim of the authors is to evaluate carefully the genes and their mutations linked   to this system,  able to modify the outer and inner bacterial membranes in order to make the bacterium susceptible to these antibiotics or on the contrary other genes and mutations involved in maintaining the membrane integrity.

The manuscript is quite interesting, well written  and clear  even though  it is too long and too specific about  molecular data. In fact I would look better at this article published in a journal specialized in molecular methods. Anyway the genetic part concerning the genes,  the mutations, the operons etc is accurately described as well as the part concerning the MIC measurements, the fluorescence intensity detected  by the luminometer and the bacterial survival rates. Perhaps I suggest the authors to try to shorten the part of the methods; for example  if these are described sufficiently in the text, they should be more concisely mentioned in the  notes or captions of the graphs without repeating  the techniques.  

The graphs reported as Fig.6 on page 13 should be modified as Fig. 5  because the real Fig.6 is reported on page 15. Yet the authors should control that all the  acronyms  reported in the text are explained at the first instance. In fact it seems that the acronyms SDS and HEPES have not been  specified.

Author Response

In this article the authors examine in detail the activity of the PhoP- PhoQ two-component regulatory system on the activity of polymyxin B and polymyxin E (colistin) against Pseudomona aeruginosa known as a very resistant microorganism for which the above antibiotics constitute the last resort. The aforementioned system has the capacity of maintaining the integrity of the cellular membrane so that the bacterial linking between the polymyxins B and E positively charged and the negatively charged phosphate groups of lipopolysaccharide (LPS) on the bacterial surface of Pseudomonas is prevented. The aim of the authors is to evaluate carefully the genes and their mutations linked to this system, able to modify the outer and inner bacterial membranes in order to make the bacterium susceptible to these antibiotics or on the contrary other genes and mutations involved in maintaining the membrane integrity.

The manuscript is quite interesting, well written and clear even though it is too long and too specific about molecular data. In fact I would look better at this article published in a journal specialized in molecular methods. Anyway the genetic part concerning the genes, the mutations, the operons etc is accurately described as well as the part concerning the MIC measurements, the fluorescence intensity detected by the luminometer and the bacterial survival rates. Perhaps I suggest the authors to try to shorten the part of the methods; for example if these are described sufficiently in the text, they should be more concisely mentioned in the notes or captions of the graphs without repeating the techniques.  

Response: We revised the figure legends to make them more concise.

The graphs reported as Fig.6 on page 13 should be modified as Fig. 5 because the real Fig.6 is reported on page 15. Yet the authors should control that all the acronyms reported in the text are explained at the first instance. In fact it seems that the acronyms SDS and HEPES have not been specified.

Response: We are sorry for the mistake. The figure number has been corrected. We added the explanations of the acronyms in the manuscript.

Reviewer 3 Report

In this manuscript, Yang et al. identified novel PhoPQ-dependent genes that are involved in polymyxin resistance in P. aeruginosa. The authors identified six five genes through ChIP-Seq and demonstrate that they are directly regulated by the PhoPQ TCS. The authors show that these genes are involved in maintaining membrane integrity which influences the ability of P. aeruginosa to adapt in the presence of polymyxins. The results shown by the author support their hypothesis and conclusions. Only, a two minor comments are listed below, which should be considered:

Comments

  1. Page 10, Figure 3. Labels on Panels B and C are incorrect. Panel C should be B. Panel B should be C.

  1. EMSA positive control: Since, purified PhoPHis was not phosphorylated for the EMSA experiments the observed gel shift may not be as significant as for phosphorylated PhoP. This could explain the varied amount shift when comparing to the gene expression levels (as seen in fig. 1). As such it would be prudent to also include a positive control DNA probe to show that the purified protein is causing a similar varied level gel shift.

Author Response

In this manuscript, Yang et al. identified novel PhoPQ-dependent genes that are involved in polymyxin resistance in P. aeruginosa. The authors identified six five genes through ChIP-Seq and demonstrate that they are directly regulated by the PhoPQ TCS. The authors show that these genes are involved in maintaining membrane integrity which influences the ability of P. aeruginosa to adapt in the presence of polymyxins. The results shown by the author support their hypothesis and conclusions. Only, a two minor comments are listed below, which should be considered: 

Comments

  1. Page 10, Figure 3. Labels on Panels B and C are incorrect. Panel C should be B. Panel B should be C. 

Response: We are sorry for the mistake. The labeling has been corrected.

  1. EMSA positive control: Since, purified PhoPHis was not phosphorylated for the EMSA experiments the observed gel shift may not be as significant as for phosphorylated PhoP. This could explain the varied amount shift when comparing to the gene expression levels (as seen in fig. 1). As such it would be prudent to also include a positive control DNA probe to show that the purified protein is causing a similar varied level gel shift.

Response: A previous study demonstrated the interaction between PhoP and the promoter region of PA14_46900 (listed in Table 2, reference #32). Thus the DNA probe corresponding to the PA14_46900 promoter region serves as a positive control. We revised the manuscript for clarification.